# Factors Associated with Refraining from Purchasing Foods Produced in Affected Areas after the Fukushima Daiichi Nuclear Power Station Accident

**DOI:** 10.3390/ijerph19063378

**Published:** 2022-03-13

**Authors:** Takumi Yamaguchi, Itsuko Horiguchi, Naoki Kunugita

**Affiliations:** 1Radiation Emergency Medicine Research Center, Nuclear Safety Research Association, 5-18-1 Shinbashi, Minato City, Tokyo 105-0004, Japan; 2The Support Center for Clinical Pharmacy Education and Research, Tokyo University of Science, 1-3 Kagurazaka, Shinjuku City, Tokyo 162-8601, Japan; 3Department of Nursing, School of Health Sciences, University of Occupational and Environmental Health, 1-1 Iseigaoka, Yahatanishi-ku, Kitakyushu, Fukuoka 807-8555, Japan; kunugita@med.uoeh-u.ac.jp

**Keywords:** reputational damage, radiation risk perception, Fukushima Daiichi nuclear power station accident, food safety, nuclear disaster

## Abstract

After the accident at the Fukushima Daiichi Nuclear Power Station, food products from the areas affected by the accident suffered reputational damage worldwide. Therefore, the present study aimed to identify the factors associated with people refraining from purchasing foods produced in affected areas to avoid radioactive materials. The study also aimed to clarify the time trends for the avoidance of foods produced in Fukushima Prefecture. We used data from “A survey on consumer awareness of reputational damages” conducted by the Japanese Consumers Customer Agency and implemented statistical analysis. Even if the year since the accident differed, “living with children”, “knowing detailed information about food inspections”, and “not being able to accept radiation risk even if the level is below the standard” were commonly associated. Not only did this study reveal that some people’s risk perceptions are fixed even when new knowledge is provided, but it also suggests that the implementation of food inspection can promote reputational damage. Additionally, the avoidance of Fukushima food products was found to decrease as time passed after the Fukushima accident. The results of this study may help develop countermeasures against reputational damage to food products after future nuclear disasters.

## 1. Introduction

On 11 March 2011, an accident occurred at the Fukushima Daiichi Nuclear Power Station, which caused a mass of radionuclides to be released into the atmosphere, contaminating the surrounding environment, including the land and ocean [1]. After the accident, the Japanese government set a provisional value for food and restricted the distribution of food products that exceeded 500 Bq/kg to reduce and prevent internal radiation exposure among the public [2]. As a result of this restriction, the internal radiation exposure of many Japanese people was suppressed [3]. However, Fukushima Prefecture products suffered reputational damage, and there has been a growing domestic push to avoid purchasing products from the prefecture [4]. Afterward, the Japanese government set a standard value for food at 100 Bq/kg to ensure safety [2], but this did not ease the trend of people refraining from purchasing food from the Fukushima Prefecture due to rumors. At present, although food products from the Fukushima Prefecture and other affected areas that exceed the standard value are extremely limited [5], nine countries, (China, Korea, Indonesia, United Kingdom, Iceland, Norway, Switzerland, Liechtenstein, and Russia) and five regions (Macau, Hong Kong, Taiwan, Polynesia, and the European Union) restrict the import of these products [6]. This cannot be based on science as the food is distributed and exported based on strict standard values [7]. The reputational damage of Fukushima products has been spreading, not only in Japan but also in the world, and it can be said that this damage is ongoing. The term “reputational damage” is defined in Japan as “economic damage caused by the fact that people consider foods, products, and land that are supposed to be safe to be dangerous and stop consuming or visiting them due to the extensive media coverage of an incident, accident, environmental contamination, or disaster” [8]. This kind of reputational damage has also occurred in Japan during the bovine spongiform encephalopathy problem, which led to the failure of US beef sales [9].

Since 2013, after the accident occurred, the Consumer Affairs Agency of Japan has been conducting “A survey on consumer awareness of reputational damage” [10] and has been investigating the trends of awareness of Fukushima products in Japan. However, this was a single-year assessment that did not look at trends over time. It has also not identified factors associated with people refraining from purchasing foods produced in affected areas to avoid radioactive materials. Therefore, we considered that, by clarifying the factors associated with the awareness and avoidance of foods containing radioactive materials immediately after the Fukushima accident to the present, it would be possible to examine future measures against reputational damage to food. This could also help develop countermeasures against reputational damage to food products after future nuclear disasters. In the present study, as well as factors associated with risk perception generally, we hypothesized that sex and family members living in the same household would be associated with refraining from purchasing foods produced in affected areas to avoid radioactive materials.

In the present study, we aimed to identify factors associated with this behavior. We also aimed to clarify the trends of these perceptions over a long period of time.

## 2. Materials and Methods

### 2.1. Study Design

This study employed a repeated cross-sectional study design.

### 2.2. Setting

These factors were identified and compared through 14 surveys conducted from February 2013 to January 2021. (February 2013, August 2013, February 2014, August 2014, February 2015, August 2015, February 2016, August 2016, February 2017, August 2017, February 2018, February 2019, January 2020, and January 2021.)

### 2.3. Participants

A total of 5126 men and women in their 20s to 60s who live in the areas most affected by the accident at the Fukushima Daiichi Nuclear Power Station (Iwate, Miyagi, Fukushima, and Ibaraki Prefectures) were included in this study. Included also was the same demographic living in the main destination prefectures for agricultural, forestry, and fishery products from the affected prefectures (Saitama, Chiba, Tokyo, Kanagawa, Aichi, Osaka, and Hyogo Prefectures).

### 2.4. Data Source

We submitted a request to the Consumer Affairs Agency of Japan to provide the raw data for the survey. The request was accepted. A summary of the survey results can be obtained from the Consumer Affairs Agency’s web page [10].

### 2.5. Question Items

The question items were developed by the Consumer Affairs Agency of Japan, aiming to “Continuously survey the reasons why consumers are refraining from purchasing agricultural, forestry, and marine products from the disaster-affected areas, and to utilize this information in future efforts to counter reputational damage and promote consumer understanding, including the content of explanations in risk communication”, [10]. Some of the question items were reduced the longer the amount of time from the accident grew.

The question items included demographic factors such as “Sex”, “Age”, “Living area”, “Marital status”, “Living with infants”, “Living with junior high school students”, “Living with high school students”, and “Living with elderly persons over 60 years old”.

The items also include knowledge about food inspection for radioactive materials, such as: “In municipalities where foods exceeding the standard values have been confirmed, the shipment, distribution, and consumption of such foods are stopped”; “Inspection of radioactive materials in food is implemented in 17 prefectures around the eastern region”; “Based on the guideline by the nuclear emergency response headquarter, inspection plans are formulated in municipalities, after which inspection is conducted”; “The results of an inspection are released on the web page of the Ministry of Health, Labor, and Welfare (MHLW)”; “In the case that the inspection result exceeds the screening level, inspection by Germanium (Ge) detector is implemented”; “In inspection plans, the contamination of agricultural land and the inspection results for food is stated”; “I do not know that inspection of food is conducted”. Furthermore, “Radiation risk perception”, and “Avoidance of purchasing food produced in Fukushima Prefecture” were included in the questionnaire.

### 2.6. Statistical Methods

Those who answered “Yes” to the question item “Do you refrain from purchasing foods produced in the affected areas to avoid radioactive materials?” were set as criterion variables. Additionally, we selected and analyzed the common question items from the first to 14th survey.

We used a chi-squared test to compare the independent categorical variables and selected the *p* < 0.1 variables as explanatory variables. After the chi-squared test, we used binomial logistic regression analysis to clarify the variables’ association with the avoidance of purchasing foods produced in the Fukushima Prefecture after the accident. A maximum likelihood method was used for variable selection. All data were statistically analyzed using SPSS Statistics (version 27.0; IBM, Tokyo, Japan). Statistical significance was set at *p* < 0.05.

Regarding the avoidance of foods produced in Fukushima Prefecture, the changes over time from the first survey to the last survey (the 14th survey) were summarized. Additionally, factors associated with people refraining from purchasing foods produced in the affected areas to avoid radioactive materials after the accident were compared between the results of the first survey and last survey.

## 3. Results

### 3.1. Chi-Squared Test for the First Study

When compared to “Refraining from purchasing foods produced in affected areas to avoid radioactive materials” (*n* = 1443), “Not refraining from purchasing foods produced in affected areas to avoid radioactive materials” (*n* = 2088) showed differences in “sex” (*p* < 0.001), “age” (*p* < 0.001), “residential area” (*p* < 0.001), “marital status” (*p* < 0.001), “living with infants” (*p* < 0.001), “living with elementary school children” (*p* = 0.016), “knowing that inspection of radioactive materials in food is implemented in 17 prefectures around the eastern region” (*p* = 0.001), “knowing that in case the inspection result exceeds the screening level, inspection by Ge detector is implemented” (*p* = 0.019), “not knowing that food inspections are conducted” (*p* = 0.048), and “risk perception” (*p* < 0.001) (Table 1).

### 3.2. Binomial Logistic Regression Analysis for Refraining from Purchasing Foods Produced in Affected Areas to Avoid Radioactive Materials of the First Study

With regard to the residential area, Miyagi Prefecture (OR, 0.526; 95% CI, 0.298–0.929; *p* = 0.027), Ibaraki Prefecture (OR, 0.322; 95% CI, 0.185–0.556; *p* < 0.001), Saitama Prefecture (OR, 0.518; 95% CI, 0.324–0.829; *p* = 0.006), Chiba Prefecture (OR, 0.475; 95% CI, 0.299–0.729; *p* = 0.002), Tokyo Prefecture (OR, 0.537; 95% CI, 0.343–0.842; *p* = 0.007), Kanagawa Prefecture (OR, 0.452; 95% CI, 0.343–0.842; *p* = 0.001), Aichi Prefecture (OR, 0.329; 95% CI, 0.204–0.529; *p* < 0.001), Osaka Prefecture (OR, 0.384; 95% CI, 0.241–0.614; *p* < 0.001), and Hyogo Prefecture (OR, 0.406; 95% CI, 0.248–0.666; *p* < 0.001) showed a protective factor against refraining from purchasing foods produced in affected areas to avoid radioactive materials. With regard to age, participants being in their 30s (OR, 1.372; 95% CI, 1.049–1.796; *p* = 0.021), 40s (OR, 1.499; 95% CI, 1.156–1.944; *p* = 0.002), 50s (OR, 1.329; 95% CI, 1.01–1.748; *p* = 0.042), and 60s (OR, 1.537; 95% CI, 1.185–1.993; *p* = 0.001) were risk factors for refraining from purchasing foods produced in affected areas to avoid radioactive materials. Furthermore, being female (OR, 1.396; 95% CI, 1.199–1.625; *p* < 0.001), living with infants (OR, 1.583; 95% CI, 1.247–2.01; *p* < 0.001), and an affirmative response to “Inspection of radioactive materials in food is implemented in 17 prefectures around eastern region” (OR, 1.432; 95% CI, 1.174–1.748; *p* < 0.001) were risk factors for refraining from purchasing foods produced in affected areas to avoid radioactive materials. Regarding risk perception, an affirmative response to “Cannot accept even less than the standard value” (OR, 4.103; 95% CI, 3.265–5.156; *p* < 0.001) was a risk factor. On the other hand, an affirmative response to “Not caring about risk” (OR, 0.31; 95% CI, 0.233–0.413; *p* < 0.001) was a protective factor against refraining from purchasing foods produced in affected areas to avoid radioactive materials (Table 2).

### 3.3. Chi-Square Test for the 14th Study

“Not refraining from purchasing foods produced in affected areas to avoid radioactive materials” (*n* = 2536) showed differences in “age” (*p* = 0.002), “residential area” (*p* = 0.006), “marital status” (*p* = 0.01), “having cohabitants” (*p* = 0.031), “living with elementary school children” (*p* < 0.001), “knowing that in municipalities where foods exceeding the standard values have been confirmed, shipment, distribution and consumption of the foods are stopped” (*p* < 0.001), “knowing that, based on the guideline from the nuclear emergency response headquarters, inspection plans are formulated in municipalities, after which inspection is conducted” (*p* = 0.001), “knowing that the results of an inspection are released on the web page of MHLW” (*p* < 0.001), “knowing that in case the inspection results exceed the screening level, inspection by Ge detector is implemented” (*p* < 0.001), “knowing that in inspection plans, contamination of agricultural land and the inspection results of food are stated” (*p* < 0.001), “not knowing that food inspections are conducted” (*p* = 0.048), and “risk perception” (*p* < 0.001) when compared to “refraining from purchasing foods produced in affected areas to avoid radioactive materials” (*n* = 728) (Table 3).

### 3.4. Binomial Logistic Regression Analysis for Refraining from Purchasing Foods Produced in Affected Areas to Avoid Radioactive Materials in the 14th Study

Regarding residential area, Saitama Prefecture (OR, 2.167; 95% CI, 1.099–4.276; *p* = 0.026), Tokyo Prefecture (OR, 2.398; 95% CI, 1.244–4.623; *p* = 0.009), and Kanagawa Prefecture (OR, 2.187; 95% CI, 1.121–4.267; *p* = 0.022) were risk factors for refraining from purchasing foods produced in affected areas to avoid radioactive materials. Furthermore, “being a widow” (OR, 4.455; 95% CI, 1.673–11.858; *p* = 0.003), “living with elementary school students” (OR, 1.738; 95% CI, 1.316–2.897; *p* < 0.001), and “cannot accept even less than the standard value” (OR, 3.307; 95% CI, 2.50–4.373; *p* < 0.001) were also risk factors for refraining from purchasing foods produced in affected areas to avoid radioactive materials. On the other hand, “not knowing that inspection of food is conducted” (OR, 0.65; 95% CI, 0.525–0.805; *p* = 0.006) was a preventive factor against refraining from purchasing foods produced in affected areas to avoid radioactive materials (Table 4).

### 3.5. Trend of Proportion in Avoidance of Foods Produced in Fukushima Prefecture

From the first study in February 2013 to the 14th study in January 2021, out of 5176 participants, 1004 (19.4%), 927 (17.9%), 792 (15.3%), 1014 (19.6%), 901 (17.4%), 590 (17.2%), 813 (15.7%), 859 (16.6%), 776 (15.0%), 683 (13.2%), 657 (12.7%), 647 (12.5%), 556 (10.7%), and 418 (8.1%) avoided foods produced in Fukushima Prefecture, respectively (Figure 1).

## 4. Discussion

In the present study, we investigated factors associated with the public’s perception regarding refraining from purchasing foods produced in affected areas to avoid radioactive materials and avoidance of foods produced in the Fukushima Prefecture and clarified the trend over time through 14 surveys.

In the first study, “Refraining from purchasing foods produced in affected areas to avoid radioactive materials” was associated with the following factors: residential area, age, sex, living with infants, knowledge about actual food inspection, and risk perception. On the other hand, in the 14th study, it was associated with the following factors: residential area, marital status, living with elementary school students, knowledge about actual food inspection, and risk perception. Even though living in prefectures other than Fukushima was regarded as a protective factor in the first study, living in prefectures other than Fukushima was regarded as a risk factor in the 14th study. Although crisis communication regarding radiation was conducted immediately after the accident, it has been revealed that most of the residents had no knowledge about radiation and were anxious about its potential impact on their lives [11]. As a result, their awareness and refraining from purchasing foods produced in affected areas to avoid radioactive materials increased. However, it is possible that with time the residents of the Fukushima Prefecture have acquired knowledge of radiation, as well as knowledge of radioactive materials in food and food inspections due to radiation risk communication initiatives [12]. On the other hand, it is conceivable that 10 years have passed without residents outside the Fukushima Prefecture gaining sufficient knowledge about radiation. Their awareness and avoidance of Fukushima food products may have become fixed as they may have not updated their knowledge about radiation. In fact, in a survey on the perception of radiation risks conducted among Tokyo residents, about half of the respondents answered that Fukushima residents would experience cancer and other adverse effects in later life, indicating that knowledge is not being disseminated [13]. Furthermore, according to a survey by the Japanese Ministry of Agriculture, Forestry, and Fisheries, 38.6% of people living in urban regions, including Tokyo, reported that they consumed organic vegetables at least once a week, a higher percentage than people living in other areas [14]. Although organic vegetables are generally known to be expensive, people living in urban regions choose organic foods because they have high levels of consciousness about the food they eat [15]. Therefore, in the present study, residents living in metropolitan areas who avoided foods produced in the Fukushima Prefecture could have avoided foods produced there because of their high consciousness of food safety.

As for the factors age and sex, in the first survey, age groups 30 and older were a risk factor for the avoidance of Fukushima food products. However, in the 14th survey, age was not a risk factor. In addition, being female was a risk factor in the first survey, but no sex association was found in the 14th survey. It has been shown that older age is associated with higher risk perception [16], and the results of the present study were consistent with these findings. Additionally, a previous study reported that females tended to have higher risk perceptions than males [17]. At the time of the first survey, the effects of radiation caused by the Fukushima accident were widely reported on TV and other media, and the accident was a matter of social concern. Therefore, it is possible that this is associated with the attribute of higher risk perception in older people and females. On the other hand, since the 14th survey was conducted 10 years after the accident, we considered that the accident was no longer a big concern, even among those who perceived the risks to be relatively high. That has likely to have contributed to the decrease in the tendency to avoid Fukushima food products.

In the first survey, “living with infants” was associated with refraining from purchasing foods produced in affected areas to avoid radioactive materials, and in the 14th survey, “living with elementary school children” was associated with the same. We considered that this result was related to the fact that the children who were infants at the time of the first survey had become elementary school students by the time of the 14th survey. It is well known that radiation has a significant impact on children, and concerns about pediatric thyroid cancer, which was reported in large numbers after the Chernobyl nuclear accident [18], were particularly high after the Fukushima accident. Therefore, it was assumed that the tendency to refrain from purchasing foods produced in affected areas to avoid radioactive materials was higher among parents who were concerned about potential adverse effects on their children.

In the 14th survey, the experience of bereavement was associated with refraining from purchasing foods produced in affected areas to avoid radioactive materials. It has been shown that spousal bereavement contributes to poor mental health [19]. In addition, the correlation between radiation risk perception and mental health has also been reported [20]. Therefore, this leads to a tendency to refrain from purchasing foods produced in affected areas to avoid radioactive materials and was related to the fact that the experience of bereavement from a spouse caused a decline in mental health and increased radiation risk perception. The fact that those who lost their spouses were not associated with refraining from purchasing foods produced in affected areas to avoid radioactive materials in the first survey is considered to be so because there were relatively more people who avoided Fukushima products several years after the Fukushima accident. Therefore, it is presumed that “those who had experienced bereavement from a spouse,” who may have perceived radiation as a high-risk, did not show a significant association.

In the first survey, “knowing about the inspection of radioactive materials in food is implemented in 17 prefectures around the eastern region” was a risk factor for the avoidance of food products made in the Fukushima Prefecture. In the 14th survey, “knowing that if inspection results exceeded screening level, inspection by Ge detector would be implemented” was a risk factor. In previous studies of risk perception, a negative correlation was reported between increased knowledge and decreased risk perception [21], but there was also a positive correlation between increased knowledge and increased risk perception [22]. The results of the present study revealed that gaining knowledge was associated with a tendency to refrain from purchasing foods produced in affected areas to avoid radioactive materials. Notably, the 14th survey revealed that “not knowing that food inspections are being conducted” was a protective factor for avoiding Fukushima food products, suggesting that continued food inspections possibly result in the food products suffering reputational damage.

Furthermore, in both the first and 14th surveys, regarding the radiation risk perception “cannot accept less than the standard value” was a risk factor for refraining from purchasing foods produced in affected areas to avoid radioactive materials. This could indicate that risk perception is not only a judgment based on scientific knowledge but is also an emotional decision.

Looking at the changes in the awareness and avoidance of Fukushima food products over time, it was clear that the percentage of the public avoiding Fukushima food products declined year by year. In Japan, the Reconstruction Agency and other organizations have been implementing measures to reduce reputational damage to Fukushima products [23]. In particular, the Ministry of the Environment has been conducting risk communication activities regarding health effects such as the citizens’ round-table conference [24]. In addition, the Food Safety Commission [25], Ministry of Health, Labor, and Welfare [26], Ministry of Agriculture, Forestry and Fishery, and the Consumer Affairs Agency [27] have been conducting risk communication activities regarding food products, and as a result, the avoidance of Fukushima products has reduced. On the other hand, it has been revealed that about 8% of the public still has an aversion to food products made in Fukushima Prefecture. Although food products made in Fukushima have gone through rigorous standard values before being released to the market, some members of the public are judging the risks based on emotions rather than scientific facts. The reason for the continued food inspections is to ensure food safety [28], even though most of the foods have not exceeded the standard values or been detected to have radioactive materials. On the other hand, it was found that knowing the details of food inspection was associated with refraining from purchasing food produced in the affected areas to avoid radioactive materials. Therefore, it is conceivable that food inspections to ensure food safety did not lead to relief among consumers. The results of the present study showed that knowledge of detailed food inspections was associated with refraining from purchasing foods produced in the affected area. Additionally, sex and family members living in the same household were also associated with this, as in our hypothesis. We highlighted the strength of the present study, i.e., the making of comparisons regarding refraining from purchasing foods produced in the affected area between three years after the Fukushima accident and 10 years after that and to confirm differences in the passage of time and associated factors. The findings of the present study are likely to help dispel ongoing reputational damage after the Fukushima accident and prevent this happening to foods during future nuclear disasters that may occur. 

This study had several limitations. First, this survey was not conducted for the whole of Japan, so it did not show trends for Japan as a whole. Second, since the present study was a panel survey rather than a survey of all residents, those who were interested in the present study responded to the survey, possibly creating a biased sample.

## 5. Conclusions

The tendency to avoid food produced in the Fukushima Prefecture has been decreasing year by year. Ten years after the accident, the tendency to avoid food produced in the Fukushima Prefecture was found to be only about 8%, suggesting that avoidance of Fukushima products in Japan had decreased. In addition, the following factors were associated with refraining from purchasing foods produced in affected areas to avoid radioactive materials in a survey conducted three years after the accident: “living in the Fukushima Prefecture”, “being over 30 years old”, “female”, “living with infants”, “knowing the details of food inspection practices” and “not being able to accept radiation risk even if the level is below the standard”. On the other hand, 10 years after the accident, the factors “living in the metropolitan area”, “having experienced bereavement”, “living with elementary school-aged children”, “knowing the details of food inspection practices”, “knowing that food inspections are conducted” and “not being able to accept radiation risk even if the level is below the standard” were risk factors. Even if the number of years since the accident differed, “living with children”, “knowing detailed information about food inspections” and “not being able to accept radiation risk even if the level is below the standard” were commonly associated. Not only did this study reveal that some people’s risk perceptions are fixed even when new knowledge is provided, but it also suggests that the implementation of food inspection itself possibly promotes reputational damage. The results of this study may help develop countermeasures against reputational damage to food products after future nuclear disasters.

## Figures and Tables

**Figure 1 ijerph-19-03378-f001:**
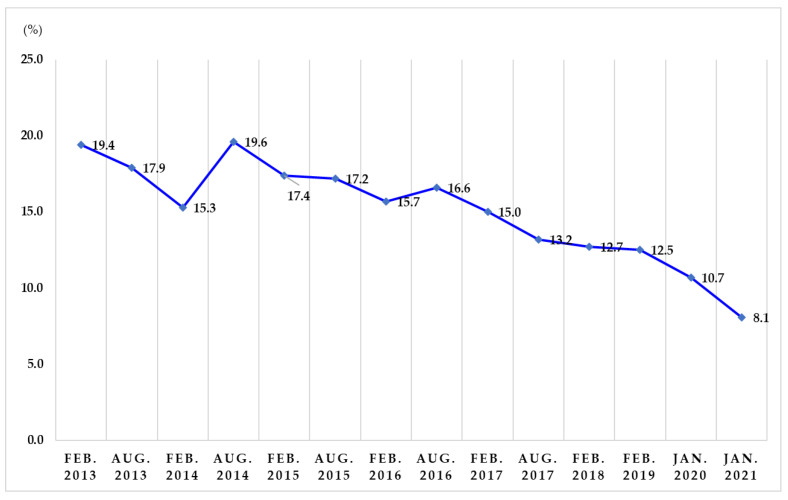
Trends in avoidance of foods produced in Fukushima Prefecture.

**Table 1 ijerph-19-03378-t001:** Comparison between refraining from purchasing foods produced in affected areas to avoid radioactive materials or not in the first survey.

		Refraining from Purchasing Foods Produced in Affected Areas to Avoid Radioactive Materials, *n* = 1443 (%)	Not Refraining from Purchasing Foods Produced in Affected Areas to Avoid Radioactive Materials, *n* = 2088 (%)	*p*-Value
Sex				
	Female	869 (60.2)	1085 (52.0)	<0.001
Age				<0.001
	20s	160 (11.1)	309 (14.8)	
	30s	356 (24.7)	415 (19.9)	
	40s	23.5 (339)	448 (21.5)	
	50s	243 (16.8)	408 (19.5)	
	60s	345 (23.9)	508 (24.3)	
Residential Area				<0.001
	Iwate	27 (1.9)	34 (1.6)	
	Miyagi	54 (3.7)	76 (3.6)	
	Fukushima	64 (4.4)	45 (2.2)	
	Ibaraki	50 (3.5)	104 (5.0)	
	Saitama	179 (12.4)	225 (10.8)	
	Chiba	135 (9.4)	201 (9.6)	
	Tokyo	315 (21.8)	390 (18.7)	
	Kanagawa	209 (14.5)	299 (14.3)	
	Aichi	135 (9.4)	258 (12.4)	
	Osaka	166 (11.5)	279 (13.4)	
	Hyogo	109 (7.6)	177 (8.5)	
Marital Status				<0.001
	Married	1041 (72.1)	1340 (64.2)	
	Unmarried	311 (21.6)	600 (28.7)	
	Divorced	58 (4.0)	110 (5.3)	
	Widowed	33 (2.3)	38 (1.8)	
Living with infants				
	Yes	256 (17.7)	219 (10.5)	<0.001
Living with elementary school students				
	Yes	181 (12.5)	207 (9.9)	0.016
Living with junior high school students				
	Yes	96 (6.7)	115 (5.5)	0.17
Living with high school students				
	Yes	107 (7.4)	137 (6.6)	0.345
Living with elderly persons over 60 years old				
	Yes	310 (21.5)	472 (22.6)	0.434
In municipalities where foods exceeding the standard values have been confirmed, shipment, distribution, and consumption of these foods stop				
	Known	926 (64.2)	1296 (62.1)	0.215
Inspection of radioactive materials in foodstuff is implemented in 17 prefectures around the eastern region				
	Known	281 (19.5)	319 (15.3)	0.001
Based on the guideline of the nuclear emergency response headquarter, inspection plans are formulated in municipalities, after which inspection is conducted				
	Known	422 (19.2)	618 (29.6)	0.851
The results of an inspection are released on the web page of MHLW				
	Known	254 (17.6)	318 (15.2)	0.063
In the case that inspection results exceed the screening level, inspection by Ge detectors is implemented				
	Known	201 (13.9)	235 (11.3)	0.019
In inspection plans, the contamination of agricultural land and the inspection results of foodstuff are stated				
	Known	287 (19.9)	372 (17.8)	0.124
I do not know that inspection of foodstuff is conducted				
	Yes	218 (15.1)	369 (17.7)	0.048
Radiation risk perception				<0.001
	He or she cannot accept less than the standard value	539 (39.0)	240 (11.7)	
	He or she can accept less than the standard value	541 (39.1)	894 (43.4)	
	He or she does not care	87 (6.3)	515 (25.0)	
	He or she cannot decide due to insufficient information	215 (15.6)	409 (19.9)	

**Table 2 ijerph-19-03378-t002:** Logistic regression analysis of refraining from purchasing foods produced in affected areas to avoid radioactive materials in the first survey.

				95% CI			
		B	OR	Lower	Upper	*p*-Value	VIF
Residential Area							1.002
	Fukushima	(Reference)					
	Iwate	−0.603	0.547	0.273	1.096	0.089	
	Miyagi	−0.642	0.526	0.298	0.929	0.027	
	Ibaraki	−1.132	0.322	0.185	0.563	<0.001	
	Saitama	−0.658	0.518	0.324	0.829	0.006	
	Chiba	−0.744	0.475	0.294	0.769	0.002	
	Tokyo	−0.621	0.537	0.343	0.842	0.007	
	Kanagawa	−0.794	0.452	0.285	0.717	0.001	
	Aichi	−1.113	0.329	0.204	0.529	<0.001	
	Osaka	−0.956	0.384	0.241	0.614	<0.001	
	Hyogo	−0.901	0.406	0.248	0.666	<0.001	
Age							1.095
	20s	(Reference)					
	30s	0.317	1.372	1.049	1.796	0.021	
	40s	0.405	1.499	1.156	1.944	0.002	
	50s	0.284	1.329	1.01	1.748	0.042	
	60s	0.43	1.537	1.185	1.993	0.001	
Sex							1.013
	Male	(Reference)					
	Female	0.334	1.396	1.199	1.625	<0.001	
Living with infants							1.103
	No	(Reference)					
	Yes	0.459	1.583	1.247	2.01	<0.001	
Inspection of radioactive materials in foodstuff is implemented in 17 prefectures around the eastern region.							1.018
	No						
	Yes	0.359	1.432	1.174	1.748	<0.001	
Radiation risk perception							1.014
	He or she cannot decide due to insufficient information	(Reference)					
	He or she cannot accept less than the standard value	1.412	4.103	3.265	5.156	<0.001	
	He or she can accept less than the standard value	0.044	1.045	0.853	1.28	0.669	
	He or she does not care	−1.17	0.31	0.233	0.413	<0.001	

**Table 3 ijerph-19-03378-t003:** Comparison between refraining from purchasing foods produced in affected areas to avoid radioactive materials or not in the 14th survey.

		Refraining from Purchasing Foods Produced in Affected Areas to Avoid Radioactive Materials, *n* = 728 (%)	Not Refraining from Purchasing Foods Produced in Affected Areas to Avoid Radioactive Materials, *n* = 2536 (%)	*p*-Value
Sex				
	Female	416 (57.1)	1417 (55.9)	0.544
Age				0.002
	20s	75 (10.3)	374 (14.7)	
	30s	141 (19.4)	558 (22.0)	
	40s	186 (25.5)	561 (22.1)	
	50s	158 (21.7)	453 (17.9)	
	60s	168 (23.1)	590 (23.3)	
Residential Area				0.006
	Iwate	9 (1.2)	45 (1.8)	
	Miyagi	22 (3.0)	94 (3.7)	
	Fukushima	15 (2.1)	79 (3.1)	
	Ibaraki	24 (3.3)	115 (4.5)	
	Saitama	83 (11.4)	265 (10.4)	
	Chiba	66 (9.1)	233 (9.2)	
	Tokyo	180 (24.7)	490 (19.3)	
	Kanagawa	111 (15.2)	341 (13.4)	
	Aichi	60 (8.2)	304 (12.0)	
	Osaka	92 (12.6)	362 (14.3)	
	Hyogo	66 (9.1)	208 (8.2)	
Marital Status				0.01
	Married	495 (68.0)	1621 (63.9)	
	Unmarried	173 (23.8)	738 (29.1)	
	Divorced	44 (6.0)	147 (5.8)	
	Widowed	16 (2.2)	30 (1.2)	
Having a cohabitant				
	Yes	628 (86.3)	2101 (64.4)	0.031
Living with infants				
	Yes	118 (18.8)	357 (17.0)	0.308
Living with elementary school students				
	Yes	98 (15.6)	216 (10.3)	<0.001
Living with junior high school students				
	Yes	58 (9.2)	151 (7.2)	0.104
Living with high school students				
	Yes	65 (10.4)	174 (8.3)	0.108
Living with elderly persons over 60 years old				
	Yes	134 (21.3)	513 (24.4)	0.121
Being and/or living with someone pregnant				
	Yes	6 (1.0)	33 (1.6)	0.338
In municipalities where foods exceeding the standard values have been confirmed, shipment, distribution, and consumption of the foodstuff are stopped				
	Known	246 (33.8)	617 (24.3)	<0.001
Inspection of radioactive materials in foodstuff is implemented in 17 prefectures around the eastern region				
	Known	120 (16.5)	258 (10.2)	<0.001
Based on the guideline from the nuclear emergency response headquarter, inspection plans are formulated in municipalities, after which inspection is conducted				
	Known	151 (20.7)	386 (15.2)	0.001
The results of an inspection are released on the web page of MHLW				
	Known	150 (20.6)	317 (12.5)	<0.001
In case the inspection results exceed the screening level, inspection by Ge detector is implemented				
	Known	101 (13.9)	170 (6.7)	<0.001
In inspection plans, the contamination of agricultural land and the inspection results of foodstuff are stated				
	Known	138 (19.0)	323 (12.7)	<0.001
I do not know that inspection of foodstuff is conducted				
	Yes	321 (44.1)	1456 (57.4)	<0.001
Radiation risk perception				<0.001
	He or she cannot accept less than the standard value	265 (37.3)	388 (15.4)	
	He or she can accept less than the standard value	239 (33.6)	1013 (40.2)	
	He or she does not care	77 (10.8)	447 (17.7)	
	He or she cannot decide due to insufficient information	130 (18.3)	672 (26.7)	

**Table 4 ijerph-19-03378-t004:** Logistic regression analysis of refraining from purchasing foods produced in affected areas to avoid radioactive materials in the 14th survey.

				95% CI			
		B	OR	Lower	Upper	*p*-Value	VIF
Residential Area							1.002
	Fukushima	(Reference)					
	Iwate	0.229	1.258	0.458	3.457	0.657	
	Miyagi	0.393	1.482	0.655	3.351	0.345	
	Ibaraki	0.258	1.294	0.583	2.872	0.526	
	Saitama	0.774	2.167	1.099	4.276	0.026	
	Chiba	0.626	1.87	0.932	3.754	0.078	
	Tokyo	0.875	2.398	1.244	4.623	0.009	
	Kanagawa	0.783	2.187	1.121	4.267	0.022	
	Aichi	0.14	1.15	0.57	2.32	0.695	
	Osaka	0.539	1.714	0.871	3.372	0.119	
	Hyogo	0.686	1.986	0.985	4.004	0.055	
Marital Status							1.015
	Married	(Reference)					
	Unmarried	−0.197	0.821	0.633	1.065	0.137	
	Divorced	−0.008	0.992	0.626	1.574	0.974	
	Widowed	1.494	4.455	1.673	11.858	0.003	
Living with elementary school students							1.016
	No	(Reference)					
	Yes	0.553	1.738	1.316	2.295	<0.001	
In case the inspection results exceed the screening level, inspection by Ge detector is implemented							1.123
	Unknown	(Reference)					
	Known	0.737	2.089	1.506	2.897	<0.001	
I do not know that inspection of foodstuff is conducted							1.205
	No	(Reference)					
	Yes	−0.431	0.65	0.525	0.805	0.006	
Radiation risk perception							1.086
	He or she cannot decide due to insufficient information	(Reference)					
	He or she cannot accept less than the standard value	1.196	3.307	2.5	4.373	<0.001	
	He or she can accept less than the standard value	−0.006	0.994	0.752	1.314	0.967	
	He or she does not care	−0.221	0.802	0.563	1.141	0.22	

## Data Availability

Not applicable.

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
