# Peer review of "Factors Associated with Refraining from Purchasing Foods Produced in Affected Areas after the Fukushima Daiichi Nuclear Power Station Accident"

_ijerph, 2022, doi:10.3390/ijerph19063378_

Round 1

Reviewer 1 Report

The paper entitled: “Factors Associated with Avoidance of Foods Produced in Fukushima Prefecture after the Fukushima Daiichi Nuclear PowerStation Accident”   aimed to identify the factors associated with the general public's awareness regarding the avoidance of foods produced in the Fukushima Prefecture and to clarify the time trends of this awareness. This is a very important issue that deserves deep reflection.

This study is original and the topic is interesting enough to attract the readers’ attention.  

This work could be published after major revision.

My observations are as follows:

  • English revision of the entire manuscript is necessary

  • Better explain the usefulness of this study and the objectives ( both in the abstract and in introduction and discussion)

  • In table 1 it is not enough clear the difference beetween Avoidance of foods produced in Fukushima Prefecture (+), and Avoidance of foods produced in Fukushima Prefecture (-). It is possible clarify this in the test?

  • factors associated with public perception and avoidance of foods produced in Fukushima prefecture by what criteria were they chosen?

  • Is a food analysis known to show that there are no alterations and contaminants so as not to justify this 8% who still do not trust? Argue in discussion

Reviewer 2 Report

Manuscript ID: ijerph-1620236

The manuscript aimed to identify factors associated with the public's perception and avoidance of foods produced in the Fukushima Prefecture and clarify the trends of these perceptions over a long period. The theme is interesting and pertinent. However, it is necessary to describe better the methods’ and results’ sections and some other reviews that I highlight below.

  • Line 44 – differentiate country/regions
  • Line 72 – change “1.st” to “1st
  • Item 2.3 – Include exclusion criteria. How did you select the age group? Insert the reference. How many subjects were able to participate in your study per year? How did you calculate the minimum sample? Did you use a systematic or convenience sample? Should (or could) the participants be the same throughout the years of studies?
  • Item 2.5 - Did you create/validate the questionnaire? Insert the information about questionnaire construction and validation. I also recommend inserting the questionnaire as supplementary material.
  • In the result section and Figure 1 title or footnote, it is important to insert the number of total respondents besides those that avoided foods produced in Fukushima Prefecture. The percentage of responses of those who avoided foods produced in Fukushima Prefecture regarding the total of respondents.
  • The Figure title should be placed below the figure.
  • Table 1 and 3 – Why did you use only 1st and 14th years? Since you have data from all years, you should use it or compare data from the 14th year with the beginning. In this sense, if you have a representative sample, you can show if the avoidance increased or decreased over the years.
  • Table 1 should be placed before Figure 1.

Thank you for the opportunity to review this manuscript!

Round 2

Reviewer 1 Report

Accept in the present form

Author Response

Dear reviewer 1

We deeply appreciate your comment. 

Reviewer 2 Report

I congratulate the authors for the manuscript improvement. I recommend only some minor reviews:

Insert the hypothesis in the introductions section and mention in the conclusion if your hypothesis was confirmed or not.

Figure 1 should be deleted from line 149 since it is placed at the end of the results section.

Highlight the strengths of your study before the study limitations.

Thank you for the opportunity to review this manuscript!
